# Single Nucleotide Polymorphisms May Increase the Risk of Aspiration Pneumonia in Post-Stroke Patients with Dysphagia

Hae-Yeon Park [1], Hyun-Mi Oh [2,3], Tae-Woo Kim [2,3], Youngkook Kim [4], Geun-Young Park [5], Hyemi Hwang [5] and Sun Im [5,*]

[1] Department of Rehabilitation Medicine, Seoul St. Mary's Hospital, College of Medicine, The Catholic University of Korea, Seoul 06591, Korea

[2] Department of Rehabilitation Medicine, National Traffic Injury Rehabilitation Hospital, Yangpyeong 12564, Korea

[3] Department of Rehabilitation Medicine, Seoul National University Hospital, Seoul 03080, Korea

[4] Department of Rehabilitation Medicine, Yeouido St. Mary's Hospital, College of Medicine, The Catholic University of Korea, Seoul 07345, Korea

[5] Department of Rehabilitation Medicine, Bucheon St. Mary's Hospital, College of Medicine, The Catholic University of Korea, Seoul 14647, Korea

* Correspondence: lafolia@catholic.ac.kr; Tel.: +82-32-340-2170

**Abstract:** This study aimed to evaluate whether genetic polymorphism is associated with an increased risk of infection, specifically post-stroke aspiration pneumonia. Blood samples were obtained from a total of 206 post-stroke participants (males, *n* = 136; mean age, 63.8 years). Genotyping was performed for catechol-O-methyltransferase (*rs4680, rs165599*), dopamine receptors (DRD1; *rs4532*, DRD2; *rs1800497*, DRD3; *rs6280*), brain-derived neurotrophic factor (*rs6265*), apolipoprotein E (*rs429358, rs7412*), and the interleukin-1 receptor antagonist gene (*rs4251961*). The subjects were stratified into two groups, aged < 65 (young) and ≥ 65 (elderly). Functional parameters and swallowing outcomes were measured at enrollment and at 3 months post-onset. The primary outcome was the incidence of aspiration pneumonia. Analysis of the association between genetic polymorphisms and aspiration pneumonia history showed that a minor C *rs429358* allele was associated with the occurrence of aspiration pneumonia in the young group, both in the additive and the dominant models (odds ratio: 4.53; 95% CI: 1.60–12.84, *p* = 0.004). In the multivariable analysis, the minor C *rs429358* allele increased the risk of post-stroke aspiration pneumonia in young stroke patients by 5.35 (95% CI: 1.64–20.88). In contrast, no such association was observed in the elderly group. Apolipoprotein E polymorphism may affect the risk of post-stroke aspiration pneumonia.

**Keywords:** deglutition; aspiration pneumonia; single nucleotide polymorphisms; stroke; dysphagia

## 1. Introduction

Aspiration pneumonia is a common medical complication after stroke, accounting for 30–50% of the stroke population. Among the many complications, aspiration pneumonia is known as one of the leading causes of increased mortality and morbidity after stroke [1–3]. An age over 65 is an independent predictor of mortality in patients hospitalized for aspiration pneumonia [4]. Due to its detrimental effect, it is important to identify its risk factors and prevent aspiration pneumonia.

Many studies have attempted to discover the clinical factors associated with aspiration pneumonia after stroke. A prospective study of 412 patients showed that an age over 65, dysarthria or aphasia, modified Rankin Scale (mRS) scores of over 4, cognitive impairment, and failure of the water-swallow test were related to an increased risk of post-stroke pneumonia [5]. The brain lesion's location is another important factor [3], with one recent study investigating the predictive factors of aspiration pneumonia according to different stroke lesions [6]. The levels of neurological disability and swallowing impairment are

also critical predictive factors. For example, low mini-mental state examination (MMSE) scores were significantly associated with aspiration pneumonia in supratentorial infarction, while low modified Barthel index (MBI) scores and aspiration signs in videofluoroscopic swallowing studies (VFSSs) and salivagrams were significant predictors of infratentorial infarction. Similarly, a study of elderly stroke patients showed that multiple previous episodes of infarctions, high National Institutes of Health stroke scale (NIHSS) scores, and masticatory muscle paralysis were associated with post-stroke aspiration pneumonia [7]. However, whether individual single nucleotide polymorphisms (SNPs) can also increase this risk and can make certain individuals more vulnerable than others to respiratory complications is yet to be proven.

Recently, studies have reported certain SNPs to be associated with infections. For instance, early after stroke, increased plasma interleukin-1 receptor antagonist (IL-1ra), an endogenous immunomodulatory cytokine encoded by IL1RN, significantly predicted post-stroke infection [8]. The minor allele of *rs4252961* in IL1RN was associated with lower plasma IL-1ra and was related to a decreased risk of infections other than pneumonia [9,10]. In another study, apolipoprotein E (APOE)-$\epsilon$4 was found to be associated with the risk of sepsis development and progression [11]. However, to our knowledge, whether genetic polymorphism influences the development of post-stroke aspiration pneumonia, one of the most common complications after stroke, has not yet been fully investigated.

Therefore, in this study, we performed a post hoc analysis of a previous prospective trial [12] to explore whether specific genes would show an increased association of aspiration pneumonia with swallowing disturbance. In a previous trial, we assessed several SNPs and followed the patients for up to three months post-stroke. For this study, we hypothesized that specific genes may be associated with aspiration pneumonia after stroke and that the effect of these SNPs may differ according to two age groups, age < 65 (young) and age $\geq$ 65 (elderly), based on the previous study that age was an independent factor predicting aspiration pneumonia [4,5]. The aims of this study were, first, to assess which SNP influenced aspiration pneumonia in these two age groups, using a multi-inheritance model, and second, to determine whether the inclusion of this SNP with other clinical factors could help to predict post-stroke aspiration pneumonia.

## 2. Methods

### 2.1. Subjects

The participants were first-time stroke patients presenting with post-stroke dysphagia symptoms, who were admitted to the departments of rehabilitation medicine in two university-affiliated hospitals from August 2018 to July 2019. The inclusion criteria were patients: (1) with a swallowing disorder, confirmed by VFSS or a fiberoptic endoscopic evaluation of swallowing (FEES), that would require the modification of diet or tube feeding; (2) who had been diagnosed with a first-ever stroke; (3) who were hospitalized for 30 days and were followed up three months after stroke onset; (4) who agreed to participate in the study. The exclusion criteria were patients: (1) who did not meet the inclusion criteria; (2) from whom it was difficult to collect blood for genetic testing;(3) with long-term neurodegenerative diseases such as Parkinson's disease, Alzheimer's disease, Guillain-Barre syndrome, and myasthenia gravis. A detailed description of the participants has been given in a previous study [12].

### 2.2. Assessment of Aspiration Pneumonia

The primary outcome of this post hoc study was the occurrence of aspiration pneumonia within the 3-month post-stroke period. The definition of aspiration pneumonia was (1) the presence of respiratory symptoms, such as purulent sputum, tachypnea, and rales; (2) leukocytosis; (3) elevated temperature; (4) gravitational segment infiltration on a chest X-ray [13].

### 2.3. Assessment of Swallowing and Functional Outcomes

Swallowing outcomes were evaluated at baseline and at 3 months post-stroke. Swallowing function was evaluated using screening tools, including the Gugging swallowing screen (GUSS) [14] and the Mann assessment of swallowing ability (MASA) [15], with lower scores indicating a more severe state. The penetration-aspiration scale (PAS) [16] and modified barium swallow impairment profile (MBSImP©) [17] were assessed via a videofluoroscopic swallowing study (VFSS) performed by a specialist, wherein the PAS indicated the presence and severity of penetration or aspiration, and MBSImP© indicated impairment of the oral or pharyngeal phase. The oral intake of liquids and food was evaluated using the functional oral intake scale (FOIS) [18]. Lastly, the eating assessment tool (EAT-10) questionnaire [19] was used to measure the quality of life associated with dysphagia.

Baseline stroke severity was evaluated by NIHSS scores [20]. MMSE [21] and Berg balance scale (BBS) [22] scores at baseline were assessed to evaluate baseline cognitive and balance impairment. The mRS [23], functional ambulatory category (FAC) [24], and MBI [25] were assessed at baseline and at 3 months post-stroke to evaluate mobility and the activities of daily living of the participants.

### 2.4. Genotyping

Genotyping was performed with whole blood obtained from the participants who agreed to blood sampling. The allelic discrimination of each SNP was performed with 2 cc of whole blood and the TaqMan SNP genotyping assays. Catechol-O-methyltransferase (*rs4680* and *rs165599*), dopamine receptor D1 (DRD1, *rs4532*), dopamine receptor D2 (DRD2, *rs1800497*), dopamine receptor D3 (DRD3, *rs6280*), brain-derived neurotrophic factor (*rs6265*), APOE (*rs429358*, *rs7412*), and the interleukin 1 receptor antagonist gene (*rs4251961*) were evaluated. APOE genotypes ($\epsilon2/\epsilon2$, $\epsilon2/\epsilon3$, $\epsilon3/\epsilon3$, $\epsilon2/\epsilon4$, $\epsilon3/\epsilon4$, $\epsilon4/\epsilon4$) were further determined by allelic combinations of *rs429358* and *rs7412* [26]. The rationale for choosing these SNPs is provided in the previous publication [12].

### 2.5. Statistical Analysis

All statistical analyses were performed using R statistical software (version 2.15.3; R Foundation for Statistical Computing, Vienna, Austria). The Shapiro–Wilk test for normality was used to evaluate the distribution of the continuous variables. The sample size estimation method was explained in a previous study [12]. As stated above, the enrolled participants were stratified into young (< 65 years old) and elderly ($\geq$ 65 years old) groups. Between-group analyses were conducted using Student's *t*-test, the Mann–Whitney test, or the chi-squared test, as appropriate. To assess the association between each SNP and aspiration pneumonia, multiple inheritance models (additive, dominant, and recessive) were used. Bonferroni's correction was applied, due to the evaluation of 8 genes, and the significance level was determined as $p < 0.05/8 = 0.006$. The continuous variables are expressed as the mean and standard deviation or median with an interquartile range, and the categorical variables are expressed as numbers with percentages.

Univariable and multivariable analyses were performed in each age group to evaluate the factors associated with aspiration pneumonia. The independent binary variables were *rs429358* (presence of the minor allele), sex (male/female), diabetes mellitus, hypertension, alcohol, smoking, MBSImp© scores (oral > 11, pharyngeal > 9, total > 22), EAT-10 (> 15), MMSE (< 18), and FAC (< 3) scores. This study was approved by the Institutional Review Board of the Medical Center (HC17TNDI0049).

## 3. Results

### 3.1. Participants

A total of 218 participants were enrolled in the study and 206 were available for analysis [12]. Table 1 represents the baseline characteristics of the patients, according to age and their history of aspiration pneumonia. In both age groups, those with a history of aspiration pneumonia had an increased risk of intubation and tracheostomy. The initial functional

and swallowing outcomes were also better in those without aspiration pneumonia history, except for MMSE scores in the elderly and the mRS scores in both groups. In the younger age group, those with aspiration pneumonia showed more likelihood of stroke recurrence.

**Table 1.** Baseline characteristics of the participants, according to age and aspiration pneumonia history.

| | Young Age (*n* = 103) | | | Old Age (*n* = 103) | | |
|---|---|---|---|---|---|---|
| | AP (+) (*n* = 46) | AP (−) (*n* = 57) | *p*-Value | AP (+) (*n* = 63) | AP (−) (*n* = 40) | *p*-Value |
| Sex (male) | 31 (37.4) | 38 (66.7) | 1.000 | 44 (69.8) | 23 (57.5) | 0.285 |
| BMI | 22.6 ± 3.4 | 23.0 ± 2.6 | 0.541 | 21.6 ± 3.7 | 22.8 ± 3.2 | 0.090 |
| Stroke type | | | 0.366 | | | 0.093 |
| infarction | 18 (39.1) | 27 (47.4) | | 40 (63.5) | 34 (85.0) | |
| hemorrhage | 25 (54.4) | 29 (50.9) | | 21 (33.3) | 6 (15.0) | |
| both | 3 (6.5) | 1 (1.7) | | 2 (3.2) | 0 (0.0) | |
| Location | | | 0.514 | | | 0.204 |
| supratentorial | 31 (67.4) | 42 (73.7) | | 43 (68.3) | 33 (82.5) | |
| infratentorial | 11 (23.9) | 13 (22.8) | | 18 (28.6) | 7 (17.5) | |
| multiple | 4 (8.7) | 2 (3.5) | | 2 (3.2) | 0 (0.0) | |
| Side | | | 0.125 | | | 0.269 |
| right | 10 (21.7) | 23 (40.4) | | 23 (36.5) | 17 (42.5) | |
| left | 23 (50.0) | 23 (40.4) | | 28 (44.4) | 20 (50.0) | |
| bilateral | 13 (28.3) | 11 (19.2) | | 12 (19.0) | 3 (7.5) | |
| Afib | 7 (15.2) | 3 (5.3) | 0.173 | 16 (25.4) | 6 (15.0) | 0.313 |
| Recur | 12 (26.1) | 4 (7.0) | 0.017 * | 16 (25.4) | 9 (22.5) | 0.922 |
| DM | 16 (34.8) | 21 (36.8) | 0.992 | 34 (54.0) | 14 (35.0) | 0.093 |
| HBP | 29 (63.0) | 40 (70.2) | 0.579 | 52 (82.5) | 28 (70.0) | 0.213 |
| Intubation | 27 (58.7) | 20 (35.1) | 0.028 * | 28 (80.0) | 7 (20.0) | 0.009 * |
| Tracheostomy | 19 (41.3) | 9 (15.8) | 0.008 * | 18 (100.0) | 0 (0.0) | 0.001 * |
| Alcohol | 14 (30.4) | 23 (40.4) | 0.403 | 18 (28.6) | 13 (32.5) | 0.839 |
| Smoking | 14 (30.4) | 18 (31.6) | 1.000 | 18 (28.6) | 12 (30.0) | 1.000 |
| | | Initial Clinical Outcomes | | | | |
| NIHSS | 15.2 ± 6.6 | 11.7 ± 7.7 | 0.014 * | 14.6 ± 7.3 | 10.1 ± 6.2 | < 0.001 * |
| MMSE | 13.0 [1.0–22.0] | 26.0 [16.0–29.0] | < 0.001 * | 14.0 [1.0–23.0] | 17.0 [12.0–24.0] | 0.063 |
| BBS | 4.0 [0.0–28.0] | 30.0 [5.0–50.0] | 0.001 * | 3.0 [0.0–8.0] | 12.5 [3.0–48.0] | 0.002 * |
| NPM at ≥ 12 weeks | 27 (58.7) | 10 (17.5) | < 0.001 * | 39 (61.9) | 4 (10.0) | < 0.001 * |
| MBSImp-Oral | 15.0 [10.0–18.0] | 8.5 [5.0–13.0] | < 0.001 * | 12.0 [10.0–16.0] | 10.0 [6.5–14.0] | 0.003 * |
| MBSImp-Pharyngeal | 11.0 [9.0–13.0] | 8.0 [5.0–13.0] | 0.024 * | 10.0 [8.0–15.0] | 6.0 [4.5–9.0] | < 0.001 * |
| GUSS | 2.0 [1.0–4.0] | 5.0 [3.0–14.0] | < 0.001 * | 2.0 [1.0–4.0] | 6.0 [3.0–14.0] | < 0.001 * |
| PAS | 8.0 [8.0–8.0] | 7.0 [6.0–8.0] | < 0.001 * | 8.0 [8.0–8.0] | 8.0 [6.0–8.0] | < 0.001 * |
| MASA | 115.0 [86.0–139.0] | 154.0 [125.0–175.0] | < 0.001 * | 119.0 [82.0–146.0] | 154.5 [139.5–173.0] | < 0.001 * |
| EAT-10 | 40.0 [40.0–40.0] | 38.0 [20.0–40.0] | 0.001 * | 40.0 [40.0–40.0] | 32.0 [20.5–40.0] | < 0.001 * |
| FOIS | 1.0 [1.0–1.0] | 1.0 [1.0–2.0] | < 0.001 * | 1.0 [1.0–1.0] | 1.0 [1.0–2.0] | 0.001 * |
| FAC | 0.0 [0.0–1.0] | 2.0 [0.0–3.0] | < 0.001 * | 0.0 [0.0–0.0] | 0.0 [0.0–3.5] | 0.007 * |
| MBI | 8.5 [0.0–44.0] | 53.0 [23.0–82.0] | < 0.001 * | 7.0 [1.5–34.0] | 42.5 [11.5–75.5] | < 0.001 * |
| mRS (≥ 3) | 46 (100) | 55 (96.5) | 0.572 | 61 (96.8) | 38 (95.0) | 1.000 |

Values are given as a number (%), means ± SD, or median [interquartile range]. Chi-squared test, Student's *t*-test, or the Mann–Whitney test was performed to compare between groups. * *p*-values < 0.05 are used for statistical significance. Abbreviations: AP, aspiration pneumonia; BMI, body mass index; Afib, atrial fibrillation; DM, diabetes mellitus; HBP, hypertension; NIHSS, National Institutes of Health stroke scale; MMSE, mini-mental state examination; BBS, Berg balance scale; NPM, nil per mouth; MBSImp, modified barium swallow impairment profile; GUSS, Gugging swallowing screen; PAS, penetration-aspiration scale; MASA, Mann assessment of swallowing ability; EAT-10, eating assessment tool; FOIS, functional oral intake scale; FAC, functional ambulatory category; MBI, modified Barthel index; mRS, modified Rankin scale.

When only participants with a history of aspiration pneumonia were evaluated, there was a higher proportion of infarction and hypertension in the elderly, compared to the young group (data not shown).

### 3.2. Genetic Polymorphism and its Association with Aspiration Pneumonia

An analysis of the association between SNP polymorphisms and aspiration pneumonia history is shown in Table 2. Three gene models (additive, dominant, and recessive genes) showed that the minor C *rs429358* allele was associated with the occurrence of aspiration pneumonia in the young group, both in the additive and dominant models (OR = 4.53; 95% CI: 1.60–12.84, *p* = 0.004). However, no such association was seen in the elderly group.

**Table 2.** Analysis of APOE4 (*rs429358*) polymorphisms and statistical association with aspiration pneumonia.

| | All Ages (*n* = 206) | | | Age < 65 (*n* = 103) | | | Age ≥ 65 (*n* = 103) | | |
|---|---|---|---|---|---|---|---|---|---|
| | **Additive OR (95% CI)** | **Dominant OR (95% CI)** | **Recessive OR (95% CI)** | **Additive OR (95% CI)** | **Dominant OR (95% CI)** | **Recessive OR (95% CI)** | **Additive OR (95% CI)** | **Dominant OR (95% CI)** | **Recessive OR (95% CI)** |
| *rs429358* | 1.92 (1.01–3.65) * | 1.87 (0.96–3.65) | N/A | 4.53 (1.60–12.84) † | 4.53 (1.60–12.84) † | N/A | 0.98 (0.43–2.21) | 0.82 (0.33–2.31) | N/A |
| *rs7412* | 0.73 (0.31–1.7) | 0.73 (0.31–1.7) | N/A | 0.99 (0.25–3.92) | 0.99 (0.25–3.92) | N/A | 0.5 (0.17–1.51) | 0.5 (0.17–1.51) | N/A |
| *rs165599* | 1.20 (0.80–1.82) | 1.30 (0.71–2.37) | 1.23 (0.59–2.56) | 1.18 (0.66–2.08) | 1.59 (0.66–3.81) | 0.88 (0.32–2.41) | 1.28 (0.69–2.36) | 1.07 (0.45–2.56) | 2.12 (0.63–7.10) |
| *rs4251961* | 0.95 (0.47–1.93) | 0.95 (0.43–2.08) | 0.89 (0.06–14.40) | 0.78 (0.29–2.07) | 0.84 (0.29–2.42) | N/A | 1.42 (0.44–4.55) | 1.31 (0.37–4.67) | N/A |
| *rs4532* | 0.92 (0.51–1.67) | 0.92 (0.47–1.79) | 0.89 (0.12–6.43) | 1.17 (0.50–2.77) | 1.32 (0.53–3.31) | N/A | 0.77 (0.34–1.75) | 0.65 (0.24–1.77) | 1.28 (0.11–14.58) |
| *rs1800497* | 1.06 (0.70–1.59) | 1.06 (0.60–1.87) | 1.11 (0.51–2.45) | 1.59 (0.88–2.89) | 1.53 (0.65–3.60) | 2.36 (0.79–7.08) | 0.77 (0.43–1.39) | 0.85 (0.38–1.93) | 0.50 (0.15–1.60) |
| *rs6280* | 1.03 (0.65–1.64) | 1.11 (0.64–1.93) | 0.73 (0.22–2.47) | 1.18 (0.61–2.28) | 1.52 (0.70–3.33) | 0.29 (0.03–2.73) | 0.84 (0.43–1.63) | 0.73 (0.33–1.64) | 1.29 (0.22–7.38) |
| *rs4680* | 0.88 (0.56–1.37) | 0.88 (0.51–1.52) | 0.75 (0.24–2.31) | 0.84 (0.46–1.53) | 0.73 (0.33–1.64) | 0.99 (0.25–3.92) | 0.86 (0.43–1.71) | 0.88 (0.40–1.94) | 0.62 (0.08–4.61) |
| *rs6265* | 1.14 (0.80–1.64) | 1.04 (0.58–1.87) | 1.44 (0.76–2.72) | 1.23 (0.73–2.10) | 1.20 (0.51–2.87) | 1.48 (0.61–3.58) | 1.14 (0.68–1.92) | 1.04 (0.46–2.37) | 1.48 (0.57–3.84) |

Major allele; M, Minor allele; m. Additive; MM (Reference) vs. Mm + 2 * mm (assumption: dose-response, MM < Mm < mm). Dominant; MM (Reference) vs. Mm + mm. Recessive; MM + Mm (Reference) vs. mm. * *p* value = 0.046; † *p*-value = 0.004. Abbreviations: OR, odds ratio; CI, confidence interval; N/A, not applicable.

### 3.3. APOE Genotyping

APOE genotyping ($\epsilon2/\epsilon2$, $\epsilon2/\epsilon3$, $\epsilon3/\epsilon3$, $\epsilon2/\epsilon4$, $\epsilon3/\epsilon4$, $\epsilon4/\epsilon4$) was performed by the combination of *rs429358* and *rs7412* [26]. No participants had the APOE $\epsilon2/\epsilon2$ genotype in this study. Those with the minor C *rs429358* allele (either CT or CC) were categorized as either $\epsilon2/\epsilon4$, $\epsilon3/\epsilon4$, or $\epsilon4/\epsilon4$ genotypes, and those with a homozygous major T allele of *rs429358* were categorized as either $\epsilon2/\epsilon3$ or $\epsilon3/\epsilon3$ genotypes, regardless of the *rs7412* allele. Therefore, those with a minor C *rs429358* allele were categorized as APOE $\epsilon4$.

### 3.4. Comparison of APOE Genotypes

The comparison of baseline parameters according to APOE genotype in the young age group showed that those with APOE $\epsilon4$ had more aspiration pneumonia and had poorer initial PAS scores (Table S1). In contrast, the proportion of patients with post-stroke aspiration pneumonia was significantly higher in those with APOE $\epsilon4$ in the young age group, and the incidence of urinary tract infections and colitis did not differ according to APOE genotypes (Figure 1). In the elderly group, no baseline functional and swallowing parameters significantly differed according to the presence of APOE $\epsilon4$ (Table S1).

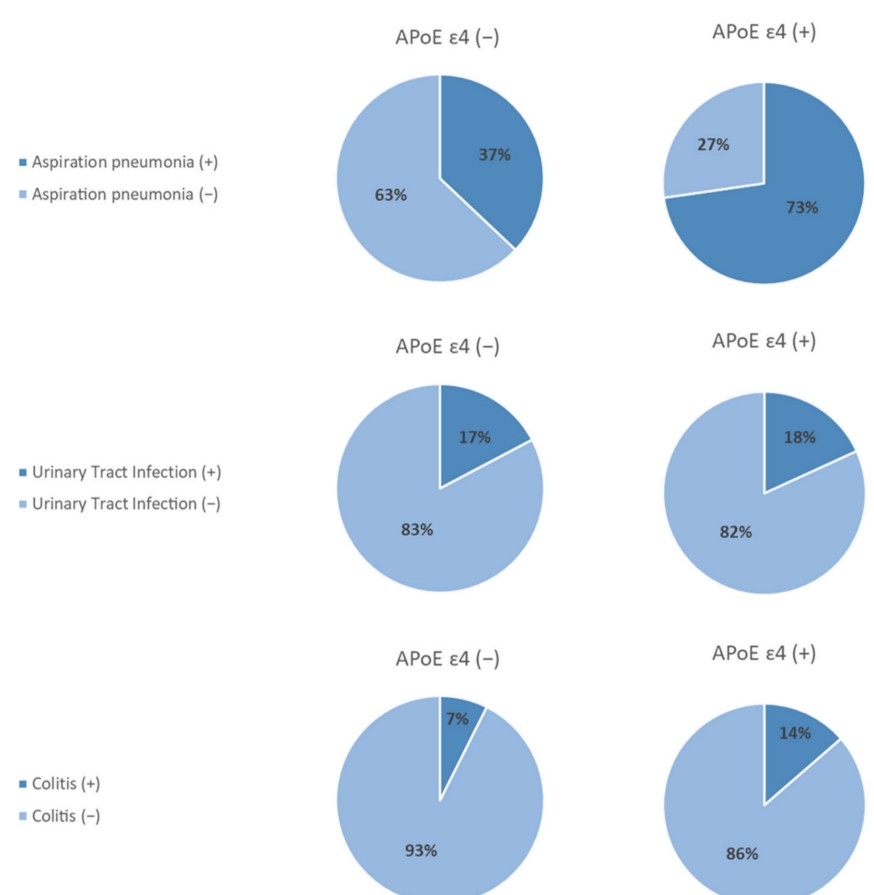

**Figure 1.** Association of infection and APOE genotypes in the young age group. The incidence of post-stroke aspiration pneumonia was significantly higher in those with APOE *ϵ*4 (*p* = 0.003). The incidence of urinary tract infections (*p* = 0.922) and colitis (*p* = 0.359) did not differ according to APOE genotypes.

The evaluation of changes in the swallowing and functional parameters from baseline to 3 months showed no statistically significant differences according to APOE genotypes, in either the young or elderly groups (Table S2).

*3.5. Univariable and Multivariable Analysis of Predictors of Aspiration Pneumonia*

The minor C *rs429358* allele (OR = 5.35; 95% CI: 1.64–20.88), poor oral MBSImP© stage score (OR = 2.73; 95% CI: 1.04–7.41), and higher mRS scores (OR = 2.45; 95% CI: 1.30–5.00) were associated with an increased occurrence of aspiration pneumonia in the young age group. In this group, a final multivariable model that included rs429358, poor oral MBSImP© stage score, worse PAS scores, and higher mRS scores, showed an area under the receiver operating characteristic curve (AUROC) of 0.82 (95% CI: 0.7–0.90) (Table 3).

In contrast, in the elderly group, *rs429358* did not significantly increase the risk of aspiration pneumonia. Instead, among the included factors, poor GUSS scores (OR = 0.80; 95% CI: 0.68–0.93) and PAS scores (OR = 1.88; 95% CI: 1.22–3.39) increased the aspiration pneumonia risk in a multivariable model with an AUROC of 0.89 (95% CI: 0.83–0.95) (Table 4).

**Table 3.** Univariable and multivariable analysis in the young age group.

| | Univariable | | Multivariable | |
|---|---|---|---|---|
| | **OR (95% CI)** | ***p*-Value** | **OR (95% CI)** | ***p*-Value** |
| *rs429358* | 4.53 (1.67–13.80) | 0.004 * | 5.35 (1.64–20.88) | 0.009 * |
| Sex (male) | 0.97 (0.42–2.21) | 0.938 | | |
| BMI | 0.98 (0.85–1.11) | 0.722 | | |
| DM (yes) | 0.91 (0.40–2.06) | 0.829 | | |
| HBP | 0.73 (0.32–1.66) | 0.445 | | |
| Alcohol (yes) | 0.65 (0.28–1.46) | 0.298 | | |
| Smoking (yes) | 0.95 (0.41–2.19) | 0.901 | | |
| MBSImp-Oral (> 11) | 4.03 (1.79–9.43) | 0.001 * | 2.73 (1.04–7.41) | 0.043 * |
| PAS | 1.92 (1.31–3.24) | 0.004 * | 1.42 (0.97–2.43) | 0.126 |
| EAT-10 (> 15) | 6.30 (1.06–120.15) | 0.091 | | |
| mRS | 2.77 (1.63–5.02) | < 0.001 * | 2.45 (1.30–5.00) | 0.009 * |
| AUROC | | | 0.82 (0.74–0.90) | |

Independent binary variables are *rs429358* (presence of minor allele), sex (male/female), diabetes mellitus, hypertension, alcohol, smoking, MBSImp scores (oral > 11), and EAT-10 (>15). * *p*-value < 0.05 is used for statistical significance. Abbreviations: OR, odds ratio; CI, confidence interval; BMI, body mass index; DM, diabetes mellitus; HBP, hypertension; MBSImp, modified barium swallow impairment profile; PAS, penetration-aspiration scale; EAT-10, eating assessment tool; mRS, modified Rankin scale; AUROC, area under the receiver operating characteristic curve.

**Table 4.** Univariable and multivariable analysis in the older age group.

| | Univariable | | Multivariable | |
|---|---|---|---|---|
| | **OR (95% CI)** | ***p*-Value** | **OR (95% CI)** | ***p*-Value** |
| *rs429358* | 0.82 (0.33–2.07) | 0.667 | | |
| Sex (male) | 0.64 (0.27–1.49) | 0.295 | | |
| BMI | 0.91 (0.81–1.02) | 0.114 | | |
| DM (yes) | 2.10 (0.92–4.90) | 0.082 | | |
| HBP (yes) | 1.82 (0.69–4.79) | 0.223 | | |
| NIHSS | 1.10 (1.03–1.17) | 0.005 * | 1.06 (0.97–1.16) | 0.188 |
| Alcohol (yes) | 0.86 (0.35–2.11) | 0.732 | | |
| Smoking (yes) | 0.89 (0.36–2.25) | 0.806 | | |
| MMSE (< 18) | 1.10 (0.49–2.49) | 0.819 | | |
| MBSImp-Oral (> 11) | 2.35 (1.03–5.57) | 0.047 * | 0.26 (0.06–0.98) | 0.062 |
| GUSS | 0.78 (0.69–0.86) | <0.001 * | 0.80 (0.68–0.93) | 0.007 * |
| PAS | 2.03 (1.36–3.38) | 0.002 * | 1.88 (1.22–3.39) | 0.013 * |
| EAT-10 (> 15) | 5.06 (0.62–104.41) | 0.168 | | |
| FOIS | 0.31 (0.11–0.61) | 0.006 * | 0.43 (0.12–1.19) | 0.150 |
| FAC (< 3) | 6.42 (2.19–21.74) | 0.001 * | 4.12 (0.94–19.96) | 0.064 |
| AUROC | | | 0.89 (0.83–0.95) | |

Independent binary variables are *rs429358* (presence of minor allele), sex (male/female), diabetes mellitus, hypertension, alcohol, smoking, MMSE (< 18), MBSImp scores (oral > 11), EAT-10 (> 15), and FAC (< 3). * *p*-value < 0.05 is used for statistical significance. Abbreviations: OR, odds ratio; CI, confidence interval; BMI, body mass index; DM, diabetes mellitus; HBP, hypertension; NIHSS, National Institutes of Health Stroke Scale; MMSE, mini-mental state examination; MBSImp, modified barium swallow impairment profile; GUSS, Gugging swallowing screen; PAS, penetration-aspiration scale; EAT-10, eating assessment tool; FOIS, functional oral intake scale; FAC, functional ambulatory category; AUROC, the area under the receiver operating characteristic curve.

## 4. Discussion

This study showed that the predictive factors for aspiration pneumonia may differ between young and elderly stroke patients, with the SNP influences manifested mainly in the former group. Among the various SNPs analyzed, only the APOE genotypes showed a positive association with the minor C *rs429358* allele, increasing the risk of aspiration pneumonia in young post-stroke dysphagia patients. With the inclusion of this SNP, a multivariable model that included worse baseline mRS and MBSImP© scores predicted the risk of aspiration pneumonia in the young group with high accuracy (AUROC = 0.82 (95% CI: 0.74–0.90)). In contrast, in the elderly group, a model that included only the clinical

factors with worse initial GUSS and PAS scores was shown to predict the risk of aspiration pneumonia (AUROC = 0.89 (95% CI: 0.83–0.95)).

The most well-known factors that can increase the risk of aspiration pneumonia after stroke are increased age, a higher degree of cognitive impairment, low levels of substance P or the use of angiotensin-converting enzyme inhibitors, increased severity of aspiration and dysphagia, and increased levels of post-stroke disability [5–7,27–29]. Our study's results were consistent with those of previous studies, reflected by the lower GUSS and higher PAS and NIHSS scores in the elderly group and the increased severity of the oral stage of impairment in the younger age stroke group with higher initial mRS scores. Of significant interest was the finding that multiple inheritance analysis showed that specific SNPs, specifically *rs4252961*, were also associated with an increased risk of aspiration pneumonia in the young age group.

One of the unexpected findings was that APOE was the SNP associated with aspiration pneumonia, which contradicts previous findings that reported that the major allele of *rs4252961* in *IL1RN* was associated with infection risk [8]. APOE is encoded by three alleles ($\epsilon$2, $\epsilon$3, and $\epsilon$4) on chromosome 19q13 and is produced by the liver, brain, spleen, kidneys, lungs, and muscle tissue [30]. By binding to lipoprotein particles and cellular receptors, APOE controls triglyceride homeostasis and lipoprotein clearance in tissues, including the lungs [31]. This regulation is affected by APOE polymorphism, and APOE $\epsilon$4 polymorphism is known to increase the risk of atherosclerosis and neurodegenerative disorders, including Alzheimer's disease and dementia. This allele can also lead to higher mortality in dementia patients and is known to be associated with younger stroke onset [32–35]. This SNP has also been related to an increased risk of swallowing dysfunction in the geriatric community population [36].

Previous studies have discussed the role of APOE genotypes in infection, especially in respiratory disease. APOE acts on macrophages and T cells, regulating the innate immune system. In one study, when injected with bacterial endotoxin, those with APOE $\epsilon$4 showed higher hyperthermia and plasma tumor necrosis factor-alpha (TNF-$\alpha$) levels [37]. APOE $\epsilon$4 has also been shown to be related to the development of sepsis [11] and enhances the attachment of *Chlamydia pneumonia* elementary bodies to host cells [38]. Furthermore, in a recent study, those with APOE $\epsilon$4 showed a higher risk of severe COVID-19 infections [39].

Another point of interest from our study was that SNPs showed different effects in young versus elderly stroke patients. Only young adults with a minor C *rs429358* allele showed an increased risk of aspiration pneumonia in multiple inheritance models, including both additive and dominant models, showing that APOE $\epsilon$4 increased the risk of aspiration pneumonia with an OR of 4.53, even after multiple Bonferroni corrections. When included in the multivariable analysis model, this minor C *rs429358* allele remained an important risk factor for aspiration pneumonia. However, the effects of this SNP on infection were only observed for aspiration pneumonia. Minor APOE alleles seemed not to exert a similar influence on urinary or gastrointestinal tract infections, both of which may increase after stroke due to immunodepression [40]. The predilection of APOE for pneumonia is consistent with past studies advocating that the role of APOE $\epsilon$4 in lung disease may be linked to an increased risk of pulmonary complications [38,39].

In contrast, no APOE allele seemed to play a role in increasing the risk of aspiration pneumonia in the elderly group. Since old age is a vital risk factor for aspiration pneumonia [4,5], genetic polymorphism influences may have been overridden by this age factor and, thus, be less influential when compared to the young age group. In fact, 44.6% of the young and 61.2% of the elderly patients had aspiration pneumonia history, with more aspiration pneumonia in the elderly group, regardless of APOE $\epsilon$4 presence.

Some limitations of the study need to be considered. First, in this study, we compared the groups based on the presence of APOE $\epsilon$4 instead of evaluating each APOE genotype. Comparison between those with APOE $\epsilon$2/$\epsilon$4, APOE $\epsilon$3/$\epsilon$4, and APOE $\epsilon$4/$\epsilon$4 was not feasible due to the limited number of participants in the APOE $\epsilon$2/$\epsilon$4 and APOE $\epsilon$4/$\epsilon$4 groups. There were only two participants with APOE $\epsilon$2/$\epsilon$4 and APOE $\epsilon$4/$\epsilon$4 each, with

most of the participants having APOE $\epsilon3/\epsilon4$ polymorphism. The two participants with APOE $\epsilon2/\epsilon4$ did not experience aspiration pneumonia. In contrast, two APOE $\epsilon4/\epsilon4$ participants had aspiration pneumonia during the follow-up period, which may support our results that APOE $\epsilon4$ may act as a risk factor for aspiration pneumonia after stroke. Secondly, recruitment was performed in only two university-affiliated hospitals, leading to a possible selection bias. In addition, the study was conducted only on post-stroke patients. Therefore, the results of the study may not be generalizable to all patients with aspiration pneumonia related to deglutition impairment caused by other neurodegenerative disorders. In the future, a multi-center prospective study with a larger number of participants of each genotype and with different etiologies is needed to further support our findings. Finally, one would also have to consider the potential bias of the aspiration pneumonia group showing higher stroke severity. However, these factors were controlled and analyzed in the multivariable analysis. Our results showed that even after considering these potential factors, the minor allele of the APOE4 proved to be a significant contributor to increasing the risk of aspiration pneumonia.

Nevertheless, the results of this study are noteworthy since, to our knowledge, this is the first study to explore the effects of genetic polymorphisms on post-stroke aspiration pneumonia. Many studies have attempted to identify factors based on clinical information [41,42] and neuroanatomical correlates [43,44] that can help determine those at an increased risk of aspiration pneumonia. The findings from this study may help provide new information to identify stroke patients at high risk of aspiration pneumonia and guide clinicians in making medical decisions to perform dysphagia screening and treatment more aggressively in those with this allele. In recognition of the detrimental consequences of aspiration pneumonia after a stroke, these preventive measures could have important clinical implications in reducing the medical comorbidities associated with pneumonia. Further detailed studies on APOE genotyping in more young stroke patients and on whether this SNP information from other possible genotypes may increase the likelihood of the early identification of those vulnerable to post-stroke aspiration pneumonia during their recovery period are warranted. The findings provided by this study may help provide new information to identify stroke patients at high risk of aspiration pneumonia.

In summary, this study suggests that APOE $\epsilon4$ may be a risk factor for post-stroke aspiration pneumonia in patients younger than 65. Considering that aspiration pneumonia may increase post-stroke mortality, this new role of APOE may be clinically relevant. Genetic polymorphisms may be taken into consideration in predicting post-stroke aspiration pneumonia.

**Supplementary Materials:** The following supporting information can be downloaded at: https://www.mdpi.com/article/10.3390/cimb44080255/s1, Table S1: Baseline parameters according to APOE genotypes in different age groups; Table S2: Association between APOE genotypes and clinical outcomes of the patients categorized by age group.

**Author Contributions:** Conceptualization, G.-Y.P. and S.I.; methodology, G.-Y.P. and S.I.; data curation, S.I. and H.-Y.P.; writing—original draft preparation, S.I. and H.-Y.P.; writing—review and editing, H.H., H.-M.O., T.-W.K. and Y.K.; project administration, G.-Y.P. and S.I.; funding acquisition, S.I. All authors have read and agreed to the published version of the manuscript.

**Funding:** This work was supported by the National Research Foundation of Korea (NRF) grant funded by the Korean government (MSIT) 2017R1C1B501792.

**Institutional Review Board Statement:** The study was approved by the institutional review board of the Catholic Medical Center (HC17TNDI0049).

**Data Availability Statement:** The data that support the findings of this study are available on request from the corresponding author. The data are not publicly available due to ethical restrictions.

**Acknowledgments:** The statistical consultation was supported by the Department of Biostatistics of the Catholic Research Coordinating Center.

**Conflicts of Interest:** The authors declare that there is no conflict of interest.

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
