# Peer review of "Single Nucleotide Polymorphisms May Increase the Risk of Aspiration Pneumonia in Post-Stroke Patients with Dysphagia"

_cimb, doi:10.3390/cimb44080255_

Round 1

Reviewer 1 Report

This research, investigating the whether genetic polymorphism is associated with post-stroke aspiration pneumonia and showing an increased risk in younger patients when the APOE É›4 allele is present, is of great interest to help predicting post-stroke aspiration pneumonia in younger patients. The data is analysed and presented in an accurate way and the limitations of the study clearly defined. Additionally, a broader study, taking into account the different possible APOE genotypes may follow as a continuation of this work. 

Author Response

REVIEWER #1

Comment 1:

This is the first study investigating the influence of genetic polymorphism on post-stroke aspiration pneumonia. The results as well as the limitations of the study are clearly displayed and future studies confirming the impact of APOE ɛ4, including more patients and taking into account the different possible genotypes may be expected.

Authors’ response/action: We appreciate the comments made from the reviewer. We have commented that more future studies are warranted considering the different types of genotypes to find those that may help identify those at high risk of post-stroke aspiration pneumonia.

Authors’ action: These comments were added to page 10 lines 348-349.

Comment 2:

English language and style are fine/minor spell check required. 

Authors’ response/action : We have made some necessary changes of the English language and style. The revised parts are shown in red color throughout the manuscript.

We appreciate your comments and hope the revised version of the manuscript is deemed suitable for publication.

Reviewer 2 Report

Dear Editor,

I reviewed the manuscript detailed below.

Relationship between Apolipoprotein E4 polymorphism and 2 aspiration pneumonia in stroke patients

The authors investigated the occurrence of aspiration pneumonia in patients with stroke with regard to selected genetic polymorphisms. One of them was identified as potential factors associated with aspiration pneumonia in the young subpopulation. Findings depicted in this study are important, the manuscript is well written. However, I have some points for revision, which might improve the quality of the manuscript.       

1.       I wondering why one of the main results is considered in the title of the manuscript. I will pronounced it in more general manner.

2.       I will address in the discussion more intensively the potential bias, that patients with pneumonia were more severe affected by the stoke event (see results table 1)

3.       Also I will expect to speculate about potential clinical implications. For instance: more preventive measures in patients with polymorphisms?

4.       I do not understand the relevance of Figure 1. Especially the colitis seem to be misplaced. Is this an issue in post stroke patients? If you consider this is of relevance please justify it in the discussion.

5.       I also would explain the rationale behind the selected gene analyses. Was this random, or by puopose?    

Author Response

REVIEWER #2

Comment 1: I am wondering why one of the main results is considered in the title of the manuscript. I will pronounced it in more general manner.

 Authors’ response/action : We appreciate the comments made by the reviewer. We agree that the title of the paper be presented in a more generalized manner therefore we have made changes to the title as follows:

Original title:

Relationship between Apolipoprotein E4 polymorphism and aspiration pneumonia in stroke patients

Revised title:

Single nucleotide polymorphisms may increase the risk of aspiration pneumonia in post-stroke patients with dysphagia.

Comment 2: I will address in the discussion more intensively the potential bias, that patients with pneumonia were more severe affected by the stoke event (see results table 1)

Authors’ response/action : We appreciate these comments. As suggested, we have discussed the potential bias posed by this factor in the discussion section.

Finally, one would also have to consider the potential bias of the aspiration pneumonia group showing higher stroke severity. However, these factors were controlled and analyzed in the multivariable analysis. Our results showed that even after taking into consideration of these potential factors, the minor allele of the APOE4 proved to be significant contributors in increasing the risk of aspiration pneumonia. (Lines 332-337, page 9, Discussion)

Comment 3:  Also I will expect to speculate about potential clinical implications. For instance: more preventive measures in patients with polymorphisms?

Authors’ response : We appreciate the comments from the reviewer.

Authors’ action: As suggested, we have added the potential clinical implications in the main manuscript. The findings provided from this study may help provide new information to identify stroke patients at high risk of aspiration pneumonia and guide in making medical de-cisions to perform dysphagia screening and treatment more aggressively in those with this allele. In consideration of the detrimental consequences of aspiration pneumonia after a stroke, these preventive measures could have important clinical implications in reducing the medical comorbidities associated with pneumonia. (lines 342-348, page 10, Discussion)

Comment 4 :   I do not understand the relevance of Figure 1. Especially the colitis seem to be misplaced. Is this an issue in post stroke patients? If you consider this is of relevance please justify it in the discussion.

Authors’ response : Thank you for your thorough review and comment. Stroke-induced immunodepression syndrome may occur after stroke, which is characterized by decreased immune response. This phenomenon may induce increased susceptibility for infections in various organs in human body, including lung, urinary tract, and intestine. Therefore, colitis is an important source of infection after stroke. Figure 1 was prepared to demonstrate that the polymorphism of APOE was related to increased post-stroke aspiration pneumonia related to dysphagia but not to other sources of infection. The possible hypothesis how this SNP only affects pulmonary infection has been discussed in the main manuscript.

Authors’ action: We have added the related reference and have modified the discussion section with the addition of a new reference.

Minor APOE alleles seemed not to exert a similar influence on urinary or gastrointestinal tract infections, both of which may increase after stroke due to immunodepression [40]. (Lines 307-309, page 9)

[40] Hoffmann, S.; Harms, H.; Ulm, L.; Nabavi, D.G.; Mackert, B.M.; Schmehl, I.; Jungehulsing, G.J.; Montaner, J.; Bustamante, A.; Hermans, M.; et al. Stroke-induced immunodepression and dysphagia independently predict stroke-associated pneumonia - The PREDICT study. J. Cereb. Blood Flow Metab. 2017, 37, 3671-3682.

Comment 5:    I also would explain the rationale behind the selected gene analyses. Was this random, or by purpose?    

Authors’ response : As stated on page 2 (lines (69-70) the results of this study were obtained from a previous trial the authors had initially performed. The publication cited as reference 12 contains the information on the justification for choosing these SNPs for analysis.

For example, APOE4 homozygosity has also been related to self-reported swallowing dysfunction in geriatric community population. The appropriate reference was added to the manuscript.

Authors’ action: We have provided this information on the main manuscript.

Page 3, line 129-130: The rationale for choosing theses SNPs is provided in the previous publication

Page 9, line 289-290: This SNP has also been related to increased risk of swallowing dysfunction in the geriatric community population [36]

[36] Mentz H, Horan M, Payton A.Ollier W, Pendleton N, Hamdy S. Homozygositiy in the ApoE 4 polymorphism is associated with dysphagic symptoms in older adults. Dis Esophagus. 2015,28:97-103.

We appreciate your comments and hope the revised version of the manuscript is deemed suitable for publication.
